# SplitMeanFlow: Interval Splitting Consistency in Few-Step Generative Modeling

## Abstract

Flow Matching has achieved prominent performance in generative modeling, yet it is plagued by high computational costs due to iterative sampling. Recent approaches such as MeanFlow address this bottleneck by learning average velocity fields instead of instantaneous velocities. However, we demonstrate that MeanFlow's differential formulation is a special case of a more fundamental principle. In this work, we revisit the first principles of average velocity fields and derive a key algebraic identity: Interval Splitting Consistency. Building on this, we propose SplitMeanFlow, a novel framework that directly enforces this algebraic consistency as a core learning objective. Theoretically, we show that SplitMeanFlow recovers MeanFlow's differential identity in the limit, thereby establishing a more general and robust basis for average velocity field learning. Practically, SplitMeanFlow simplifies training by eliminating the need for JVP and enables one-step synthesis. Extensive experiments on large-scale speech synthesis tasks verify its superiority: SplitMeanFlow achieves a $10\times$ speedup and a $20\times$ reduction in computational cost, while preserving speech quality, delivering substantial efficiency gains without compromising generative performance.

## 1 Introduction

Generative modeling has seen remarkable advancements, with approaches such as Diffusion Models Sohl-Dickstein et al. (2015) and Flow Matching Liu et al. (2023); Song et al. (2021) establishing new benchmarks for generating high-fidelity samples across various domains, including images Rombach et al. (2022); Esser et al. (2024), video Kong et al. (2024); Bar-Tal et al. (2024), and audio Guan et al. (2024); Tian et al. (2025). Despite their impressive capabilities, these models are often limited in practical use by a significant computational bottleneck: their dependence on iterative sampling, which typically requires tens or even hundreds of neural network inferences. This high computational cost presents a major challenge, especially in resource-constrained or latency-sensitive environments. As a result, research has increasingly focused on developing "few-step" or even "one-step" generative models to mitigate this sampling overhead.

In response to this challenge, several innovative approaches have emerged. Consistency Models Song et al. (2023); Luo et al. (2023a); Peng et al. (2025); Lu & Song (2025); Liu et al. (2024); Wang et al. (2024); Luo et al. (2023b); Xiao et al. (2023), for example, introduced a novel training paradigm that enforces output consistency along the same trajectory, yielding promising results for few-step generation. Building on this, MeanFlow Geng et al. (2025) provided a significant conceptual breakthrough: for large-step generation, directly modeling the average velocity across the entire path from noise to data is more effective than modeling the instantaneous velocity at each individual point. This shift from a local to a global perspective is better suited for few-step, large-stride predictions, and has propelled MeanFlow to achieve state-of-the-art performance.

The success of MeanFlow Geng et al. (2025) has demonstrated the effectiveness of learning the average velocity field to enable few-step generation. However, the approach introduces significant challenges, both computational and practical. The reliance on a differential formulation not only results in computational inefficiencies due to the complexity of the differential operations, but also complicates implementation. Specifically, the need to support Jacobian-vector products (JVP) in custom operators makes the training of existing models under MeanFlow objectives non-trivial.

In this work, we introduce SplitMeanFlow, a new framework that addresses these issues by returning to the first principles of average velocity, which is defined as the integral of instantaneous velocity over a time interval. Unlike MeanFlow, which approximates this relationship using differential identities, SplitMeanFlow avoids derivatives and instead leverages the algebraic structure of the integral. A key insight is the additivity property of definite integrals, which allows the decomposition of the integral over an entire time interval into the sum of integrals over subintervals. This leads to a novel, purely algebraic identity, Interval Splitting Consistency, that governs the relationship between average velocities across different time intervals. We formally demonstrate that the differential identity central to MeanFlow is recovered as a special case of our formulation when the splitting point approaches the time interval endpoint. In this limit, our algebraic relation simplifies to the differential form, revealing that MeanFlow's training objective is a specific instance of our broader consistency principle. By enforcing the algebraic consistency directly as a training objective, SplitMeanFlow provides a more efficient and practical method for learning the average velocity field.

Our contributions are as follows: (1) We introduce SplitMeanFlow, a method grounded in Interval Splitting Consistency, which extends the principles of MeanFlow by providing a more general framework for learning average velocity fields. (2) SplitMeanFlow eliminates the need for JVP operations, leading to improved computational efficiency, simplified implementation, and more stable training. (3) We demonstrate the practical impact of SplitMeanFlow through its successful deployment in large-scale industrial applications, and its integration with open-source models, showcasing both its robustness and scalability.

## 2 Related Works

### 2.1 Diffusion Models and Flow Matching

Diffusion models Sohl-Dickstein et al. (2015); Song & Ermon (2019); Ho et al. (2020); Song et al. (2021); Nichol & Dhariwal (2021); Rombach et al. (2021); Peebles & Xie (2023) have achieved impressive results across various generative tasks by transforming noise into data through iterative denoising. Despite their success, these models often require hundreds of sampling steps, making inference computationally expensive and limiting their applicability in real-time settings Yang et al. (2023). To address this inefficiency, flow matching Lipman et al. (2023); Karras et al. (2022); Albergo et al. (2023) has been proposed as an alternative framework that directly learns a time-dependent velocity field to match the probability flow of a diffusion process. Flow Matching offers advantages such as faster inference and empirical performance gains. Building on this framework, recent variants such as Rectified Flow Liu et al. (2023) aim to improve training stability and convergence by adjusting the reference interpolation path. However, despite these improvements, most FM-based approaches still rely on multi-step integration during sampling. To reduce the computational burden of iterative sampling, recent research has focused on few-step generative models that aim to accelerate inference while preserving the high sample quality of diffusion-based methods.

### 2.2 Few-Step Generative Models

**Consistency Models.** Consistency models Song et al. (2023); Heek et al. (2024); Lu & Song (2024) have been developed to achieve few-step generation for visual Luo et al. (2023a); Oertell et al. (2024) and audio Fei et al. (2024); Liu et al. (2024). These methods enforce self-consistency by requiring that predictions remain invariant under repeated model application or temporal interpolation across varying noise levels. Such constraints encourage the generative trajectory to become coherent and predictable, thereby allowing accurate approximation with substantially fewer steps. Despite their empirical effectiveness, these consistency constraints are generally heuristic in nature, introduced as external regularization without explicit theoretical grounding.

**MeanFlow** MeanFlow Geng et al. (2025) presents a principled framework for one-step generative modeling by introducing the concept of *average velocity*, defined as the displacement over a time interval divided by its duration. In contrast to Flow Matching, which models instantaneous velocity at each time step, MeanFlow adopts average velocity as the learning target. It further derives an analytic relation, termed the *MeanFlow Identity*, that connects the average and instantaneous velocities via a time derivative. This formulation offers a well-grounded training objective that avoids heuristic consistency constraints and provides a clear physical interpretation.

## 3 BACKGROUND

To properly contextualize our proposed SplitMeanFlow, we first revisit the foundational principles of Flow Matching, the framework upon which our work is built. Flow Matching offers a powerful and intuitive paradigm for generative modeling, designed to learn a velocity field that transports samples from a simple prior distribution (e.g., a Gaussian) to a complex target data distribution. Both the preceding MeanFlow model and our SplitMeanFlow are fundamentally grounded in the core mechanics of this approach.

### 3.1 FLOW PATHS AND INSTANTANEOUS VELOCITY

The central idea of Flow Matching Sohl-Dickstein et al. (2015); Song & Ermon (2019); Ho et al. (2020); Song et al. (2021); Nichol & Dhariwal (2021); Rombach et al. (2021); Peebles & Xie (2023) is to define a continuous-time flow path, denoted by $z_t$, that connects a prior sample $\epsilon \sim p_{\text{prior}}(\epsilon)$ to a data sample $x \sim p_{\text{data}}(x)$. This path is typically parameterized over the time interval $t \in [0, 1]$ as:

$$z_t = a_t x + b_t \epsilon, \tag{1}$$

where $a_t$ and $b_t$ are predefined scalar schedules satisfying boundary conditions such as $a_0 = 1, b_0 = 0$ and $a_1 = 0, b_1 = 1$. A common and simple choice is the linear schedule $a_t = 1 - t$ and $b_t = t$, which defines a straight-line trajectory from $x$ at $t = 0$ to $\epsilon$ at $t = 1$.

Associated with this path is an instantaneous velocity field $v_t$, defined as the time derivative:

$$v_t = \frac{dz_t}{dt} = a'_t x + b'_t \epsilon. \tag{2}$$

Since this velocity is defined conditioned on a specific data sample $x$, it is referred to as the conditional velocity, denoted $v_t(z_t|x)$. For the linear schedule mentioned above, the conditional velocity takes the simple form $v_t = \epsilon - x$.

### 3.2 CONDITIONAL FLOW MATCHING LOSS

In practice, any given point $z_t$ on a trajectory could have been generated by numerous different $(x, \epsilon)$ pairs. The ultimate goal of a generative model is therefore not to learn any single conditional velocity, but rather the expectation over all possibilities, known as the marginal velocity $v(z_t, t) = \mathbb{E}[v_t(z_t|x)|z_t]$. However, directly computing a loss against this marginal velocity is intractable.

To circumvent this challenge, Lipman et al. (2023); Karras et al. (2022); Albergo et al. (2023) introduces an elegant and practical objective: the Conditional Flow Matching (CFM) loss. This objective trains a neural network $v_\theta$ by minimizing the discrepancy between its prediction and the easily computable conditional velocity. The loss function is formulated as:

$$\mathcal{L}_{\text{CFM}}(\theta) = \mathbb{E}_{t,x,\epsilon} \left\| v_\theta(z_t, t) - (a'_t x + b'_t \epsilon) \right\|^2. \tag{3}$$

It has been shown that minimizing this conditional loss is equivalent to minimizing the loss with respect to the true marginal velocity field. By optimizing this objective, the network $v_\theta$ effectively learns the vector field that governs the transformation of the entire distribution.

Once the model $v_\theta$ is trained, new samples can be generated by solving the ordinary differential equation (ODE) $\frac{dz_t}{dt} = v_\theta(z_t, t)$, starting from a prior sample $z_1 = \epsilon$ and integrating backward in time to $t = 0$. This integration typically requires a numerical ODE solver, which often involves multiple evaluation steps and has motivated the research into more efficient few-step and one-step generation methods.

## 4 METHOD

### 4.1 MODELING AVERAGE VELOCITY FOR EFFICIENT GENERATION

Efficiently transforming a simple prior distribution into a complex data distribution with minimal computational overhead is a fundamental goal in generative modeling, especially in few-step and one-step generation. While models like Diffusion Models and Flow Matching have made significant

progress, they face limitations when confronted with the demands of extreme efficiency. Addressing this challenge requires rethinking what a model must learn to achieve both efficiency and accuracy in generation.

Traditional generative frameworks learn an instantaneous velocity field, $v(z_t, t)$, describing the rate of change at a specific moment in time. During inference, the model simulates the trajectory from $z_1$ to $z_0$ by numerically integrating this field, solving the ordinary differential equation:

$$z_0 = z_1 - \int_0^1 v(z_\tau, \tau)d\tau. \tag{4}$$

With sufficient function evaluations, numerical solvers can generate high-quality samples. However, as the number of sampling steps decreases for efficiency, the error in numerical integration increases. In one-step generation, this reduces to a single Euler step:

$$z_0 \approx z_1 - 1 \cdot v(z_1, 1). \tag{5}$$

This approximation equates the average rate of change over the interval with the instantaneous velocity at the terminal point, introducing significant discretization error.

To achieve efficient and accurate generation, the model must learn the average velocity field, $u$, which satisfies the exact relation:

$$z_0 = z_1 - u(z_1, 0, 1). \tag{6}$$

By learning $u$, the integration burden is shifted from inference to training, eliminating discretization error by design. Modeling average velocity thus enables more efficient and accurate few-step and one-step generation.

Formally, the average velocity $u$ is defined as the integral of the instantaneous velocity $v(z_\tau, \tau)$ over the interval $[r, t]$, normalized by $t - r$. This defines $u$ as a ground-truth field, determined by the underlying instantaneous velocity field $v$, and serves as an ideal target for modeling.

However, the integral is computationally intractable during training, as it requires knowledge of the instantaneous velocity at all intermediate points along an unknown trajectory, making it unsuitable as a direct learning objective.

Given that $u$ is the ideal target but its integral is not directly computable, a tractable objective function must be designed to enable a neural network $u_\theta$ to approximate $u$. This motivates the development of two distinct methodological approaches, which are detailed in the subsequent section.

## 4.2 SPLITMEANFLOW: FROM DIFFERENTIAL TO ALGEBRAIC SOLUTION

To address the challenge of training a neural network to learn the integral-defined average velocity, two approaches arise. The first, exemplified by MeanFlow Geng et al. (2025), is a "differential solution" that bypasses direct integration by relating the average velocity $u$ to the instantaneous velocity $v$ through a differential equation. Instead of directly computing $u \propto \int v d\tau$, MeanFlow derives a differential identity called the MeanFlow Identity:

$$u(z_t, r, t) = v(z_t, t) - (t - r)\frac{d}{dt}u(z_t, r, t) \tag{7}$$

Thus, MeanFlow solves the problem indirectly by fitting the model to a differential equation derived from the integral's definition.

The second path, as proposed in this work, is the algebraic solution of SplitMeanFlow. SplitMeanFlow leverages the algebraic property of integratio, additivity, instead of relying on differentiation. The core idea is that the intrinsic structure of the integral itself provides sufficient self-consistency, eliminating the need for external differential operators. The fundamental property of additivity states

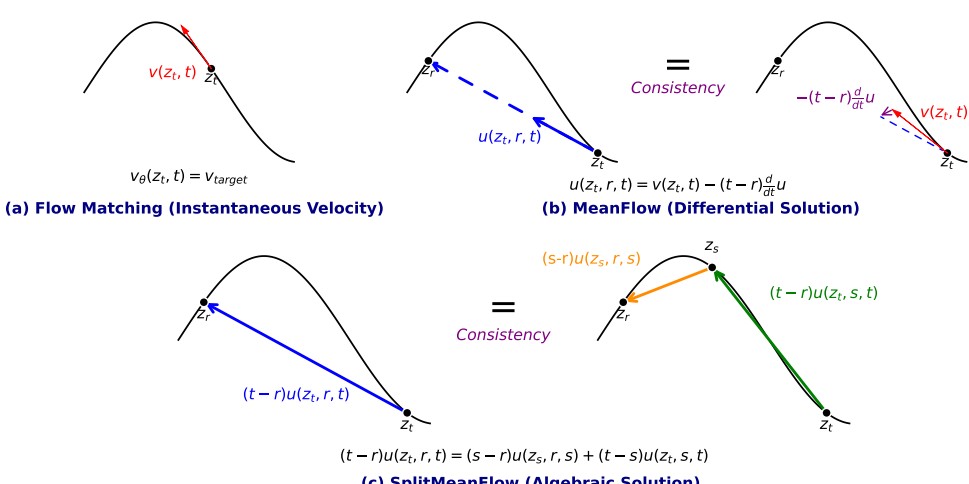

Figure 1: Conceptual Comparison of Generative Flow Methods

that the integral of a function over a full interval is the sum of the integrals over its subintervals. For any ordered time points $r \le s \le t$, this property can be expressed as:

$$\int_r^t v(z_\tau, \tau)d\tau = \int_r^s v(z_\tau, \tau)d\tau + \int_s^t v(z_\tau, \tau)d\tau \tag{8}$$

This identity reflects a key geometric property of the flow: the total displacement from time $r$ to $t$ is the sum of displacements from $r$ to $s$, and from $s$ to $t$.

To connect this principle to the average velocity $u$, we recall the definition of displacement as the product of average velocity and time duration:

$$D(a, b) = (b - a)u(z_b, a, b) = \int_a^b v(z_\tau, \tau)d\tau \tag{9}$$

This results in a purely algebraic relationship involving the average velocity field, the Interval Splitting Consistency identity:

$$(t - r)u(z_t, r, t) = (s - r)u(z_s, r, s) + (t - s)u(z_t, s, t) \tag{10}$$

This identity serves as the foundation for SplitMeanFlow, providing a self-consistent constraint on the structure of the average velocity field.

To make the training process feasible, we divide both sides of the equation by $t - r$ and define $\lambda = \frac{t-s}{t-r} \in [0, 1]$, where $s = (1 - \lambda)t + \lambda r$. This leads to the following form:

$$u(z_t, r, t) = (1 - \lambda)u(z_s, r, s) + \lambda u(z_t, s, t) \tag{11}$$

This equation can be interpreted as stating that the average velocity over the interval $[r, t]$ is a weighted sum of the average velocities over the subintervals $[r, s]$ and $[s, t]$, with the weights proportional to the length of the intervals. The formulation requires appropriate boundary conditions to ensure well-posedness and prevent degenerate solutions. Specifically, the boundary condition $u(z_t, r, t) = v(z_t, t)$ must hold when $r = t$, ensuring that the average velocity matches the instantaneous velocity at the terminal time. The full training procedure, which incorporates these boundary conditions, is outlined in Algorithm 1. Additional details on the boundary conditions are provided in the section 5.1.

---

**Algorithm 1** SplitMeanFlow Training

---

**Require:** Neural network $u_\theta$, a batch of data $x$.
1: Sample time points $r, t$ such that $0 \le r \le t \le 1$
2: Sample $\lambda \sim \mathcal{U}(0,1)$, set $s = (1-\lambda)t + \lambda r$.
3: Sample prior $\epsilon \sim \mathcal{N}(0, I)$.
4: Construct flow path point at time $t$: $z_t = (1-t)x + t\epsilon$.
5: **if** r = t **then**
6:     $target = v(z_t, t)$                                             ▷ Apply boundary condition
7: **else**
8:     $u_2 = u_\theta(z_t, s, t)$
9:     $z_s = z_t - (t-s)u_2$
10:     $u_1 = u_\theta(z_s, r, s)$
11:     $target = (1-\lambda)u_1 + \lambda u_2$
12: **end if**
13: $\mathcal{L} = \|u_\theta(z_t, r, t) - \text{sg}(target)\|$                      ▷ sg is the stop gradient function.
14: Update $\theta$ using the gradient of $\mathcal{L}$.

---

### 4.3 SPLITMEANFLOW AS A GENERALIZED CASE OF MEANFLOW

The algebraic identity at the core of SplitMeanFlow, Eq. 10, provides not only a self-contained training principle but also a theoretical generalization of the differential identity used in MeanFlow. Define $g(s) := (s-r)\, u(z_s, r, s)$, Eq. 10 can be rewritten as:

$$\frac{g(t) - g(s)}{t - s} = u(z_t, s, t). \tag{12}$$

Taking the limit $s \to t$ yields:

$$g'(t) = v(z_t, t). \tag{13}$$

Differentiate $g(t) = (t-r)\, u(z_s, r, s)$ to evaluate $g'(t)$:

$$g'(t) = u(z_t, r, t) + (t-r)\frac{d}{dt}u(z_t, r, t). \tag{14}$$

With Eq. 13 and Eq. 14 , we can recover the MeanFlow identity (Eq. 7). For a more detailed derivation of the steps leading to this conclusion, see Appendix A. This formally establishes the MeanFlow objective as a limiting special case of the SplitMeanFlow objective. As such, SplitMeanFlow serves as a more general and foundational framework that remains valid across finite intervals without resorting to infinitesimal approximations.

It is important to note that while the shortcut model Frans et al. (2025) achieves partial equivalence to our formulation for the special case $s = \frac{r+t}{2}$, the design philosophies differ significantly. SplitMeanFlow leverages the concept of average velocity and the additivity of integration to construct the identity in Eq. 10, which holds for arbitrary $s \in [r, t]$. Our method generalizes the concept of average velocity for any time points $r$, $s$, and $t$, with $0 \le r \le s \le t \le 1$, using continuous parameters. In contrast, the shortcut model uses a discrete variable.

## 5 EXPERIMENTS

**Baselines**   The proposed SplitMeanFlow method is applicable to a variety of architectures that utilize flow-based modules. To evaluate its effectiveness, we validate SplitMeanFlow on the text-to-speech (TTS) task using both open-source and private TTS models. Specifically, we conduct experiments on the following four models: (1) **F5-TTS** Chen et al. (2024), a pure flow-based non auto-regressive (NAR) TTS model that directly maps text to mel-spectrograms. An official BigV-GAN Lee et al. checkpoint is used to reconstruct the waveform. Since F5-TTS employs only flow for TTS, it serves as the primary model for our main experiments. (2) **CosyVoice2** Du et al. (2024), which first uses a language model to map text to semantic tokens, and then applies a flow-based module to map these tokens to mel-spectrograms. An official HiFiGAN Kong et al. (2020) checkpoint is used to reconstruct the waveform. (3) **DiTAR** Jia et al. (2025), which uses a language model

(LM) to convert text into continuous-space embeddings, and then maps these embeddings to the latent space of a variational autoencoder (VAE) using a flow module. (4) **Seed-TTS** Anastassiou et al. (2024), which is similar to CosyVoice2 but maps semantic tokens to VAE latents.

**Datasets**  We conduct experiments using the Emilia He et al. (2024) dataset for F5-TTS and CosyVoice2. Emilia is a multilingual and diverse in-the-wild speech dataset designed for large-scale speech generation, and for this study, we use the English and Chinese subsets, each containing 50,000 hours of speech data. For CosyVoice2, we only train the flow module, while utilizing the official checkpoint for LM. The evaluation is carried out using two benchmarks: (1) **SeedTTS test-en**, a test set from Seed-TTS containing 1,000 samples extracted from the Common Voice dataset Ardila et al. (2020), and (2) **SeedTTS test-zh**, a test set from Seed-TTS containing 2,000 samples extracted from the DiDiSpeech dataset Guo et al. (2021) for Chinese speech. Experiments on the private models are conducted with proprietary datasets.

**Metrics**  We adopt five metrics to evaluate the performance of our method: (1) Word Error Rate (WER), which measures the accuracy of speech content generation. For English, we use Whisper-large-v3 Radford et al. (2023), and for Chinese, we use Paraformer-zh Gao et al. (2023). (2) Speaker Similarity (SIM), which evaluates the similarity between the generated and reference speaker's speech using cosine similarity between speaker embeddings extracted with WavLM-large Chen et al. (2022). (3) Comparative Mean Opinion Score (CMOS), which is collected through human evaluations, with a score ranging from -2 to +2, where a higher score indicates preference for the proposed model. All CMOS evaluations in this work are compared against the corresponding flow matching baseline in each table. (4) UTMOS Saeki et al. (2022), which is evaluated using an open-source MOS prediction model, providing an estimate of human preference when exhaustive CMOS evaluations are not feasible. (5) Real-Time Factor (RTF), which measures the efficiency by calculating the time taken to generate speech relative to the input duration. For TTS models with multiple components, RTF for the flow module is reported.

## 5.1 TRAINING DETAILS

**Initialization and Distillation with Flow Matching**  While SplitMeanFlow can be trained from scratch, we find that a two-stage approach, combining pretraining and distillation, yields significantly faster convergence and superior final performance, especially for large-scale industrial applications. This strategy ensures that the SplitMeanFlow model learns from a stable and high-quality supervision signal. The first stage is dedicated to training a standard flow matching model, which will serve as the teacher. The model is trained using the standard flow matching loss, which is equivalent to our framework when the boundary condition is exclusively enforced. In the second stage, we train our SplitMeanFlow model, referred to as the student with initialization and boundary condition supervision from teacher. The detailed initialization strategy and code snippets are available in Appendix B. During distillation, the CFG Ho & Salimans (2022) dropout for the student model is set to 0.0. The teacher generates instantaneous velocity with a fixed CFG scale, enabling the CFG-free inference for $\mathcal{M}_{\text{student}}$. More details on training can be found in Appendix E

**Boundary Conditions**  The loss requires a boundary condition to avoid collapsing to a trivial solution. This anchor to reality is the instantaneous velocity condition: $u(z_t, t, t) = v(z_t, t)$. In Stage 2, same as MeanFlow, we use a flow ratio $p$ to create a mixed objective. For a fraction $p$ of the samples in a batch, we set $r = t$ to enforce the boundary condition using the teacher's velocity $v(z_t, t)$ as the target. For the remaining $1 - p$ fraction, we enforce the Interval Splitting Consistency loss(Eq. 10). However, in a distillation setup, we can adopt an approximated teacher's average velocity from $t$ to $r$ as the anchor. The details of using such an anchor are discussed in Appendix C.

## 5.2 SPLITMEANFLOW IN NAR TTS

We apply SplitMeanFlow to F5-TTS, a pure flow-based TTS model that directly converts text to mel-spectrograms. Given that F5-TTS is entirely flow-based, it benefits significantly from the acceleration introduced by SplitMeanFlow. F5-TTS necessitates the learning of an inherent time alignment between text and mel-spectrograms, which is a more challenging task compared to AR methods that leverage an LM to generate time-aligned semantic tokens. As shown in Table 1, SplitMean-

Table 1: Comparing with Flow Matching with SplitMeanFlow on F5-TTS.

| Method | NFE | CFG | WER (%, ↓) | SIM (↑) | UTMOS(↑) | CMOS (↑) | RTF(↓) |
|---|---|---|---|---|---|---|---|
| **Seed-TTS *test-en*** | | | | | | | |
| Human | N/A | N/A | 2.14 | 0.73 | 3.52 | -0.17 | N/A |
| Flow Matching | 32 | Y | 1.87 | 0.67 | 3.70 | 0.00 | 0.243 |
| SplitMeanFlow | 3 | N | 1.67 | 0.65 | 3.72 | -0.03 | 0.021 |
| SplitMeanFlow | 4 | N | 1.60 | 0.66 | 3.75 | 0.00 | 0.027 |
| **Seed-TTS *test-zh*** | | | | | | | |
| Human | N/A | N/A | 1.25 | 0.76 | 2.78 | -0.35 | N/A |
| Flow Matching | 32 | Y | 1.52 | 0.76 | 2.96 | 0.00 | 0.243 |
| SplitMeanFlow | 3 | N | 1.66 | 0.74 | 3.01 | -0.01 | 0.021 |
| SplitMeanFlow | 4 | N | 1.61 | 0.75 | 3.07 | -0.01 | 0.027 |

Flow enables 3 to 4-NFE inference on F5-TTS without compromising the quality of the generated speech, as evaluated on both the Seed-TTS English and Chinese test sets. Note that training F5-TTS with MeanFlow is not feasible due to excessive memory consumption from JVP calculations, which forces the batch size to 1. Detailed results on F5-TTS with more hyperparams setups and inference NFEs are discussed in Appendix D.1 and Appendix D.2, respectively.

## 5.3 SplitMeanFlow in AR TTS

Table 2: Comparing with Flow Matching with SplitMeanFlow on CosyVoice2.

| Method | NFE | CFG | WER (%, ↓) | SIM (↑) | UTMOS(↑) | RTF(↓) |
|---|---|---|---|---|---|---|
| **Seed-TTS *test-en*** | | | | | | |
| Human | N/A | N/A | 2.14 | 0.73 | 3.52 | N/A |
| Flow Matching | 32 | Y | 2.57 | 0.65 | 3.70 | 0.510 |
| MeanFlow | 1 | N | 2.53 | 0.64 | 3.37 | 0.026 |
| SplitMeanFlow | 1 | N | 2.49 | 0.64 | 3.71 | 0.026 |
| SplitMeanFlow | 2 | N | 2.42 | 0.65 | 3.73 | 0.050 |
| **Seed-TTS *test-zh*** | | | | | | |
| Human | N/A | N/A | 1.25 | 0.76 | 2.78 | N/A |
| Flow Matching | 32 | Y | 1.47 | 0.73 | 2.96 | 0.510 |
| MeanFlow | 1 | N | 1.66 | 0.74 | 2.78 | 0.026 |
| SplitMeanFlow | 1 | N | 1.66 | 0.74 | 2.96 | 0.026 |
| SplitMeanFlow | 2 | N | 1.61 | 0.75 | 2.98 | 0.050 |

We conduct experiments by applying SplitMeanFlow to CosyVoice2, DiTAR, and Seed-TTS. These models indirectly benefit from the flow module acceleration, as they first use an AR LM module to generate time-aligned representations, which are then mapped to mel-spectrograms or VAE latents by the flow module. This setup eliminates the need for the flow module to learn time alignment. As shown in Table 2, SplitMeanFlow achieves 1 to 2-NFE inference on CosyVoice2 without a noticeable drop in speech quality. We also conduct experiments with MeanFlow, reducing the batch size to 0.1 of the original to accommodate the memory consumption. Experiments show that while MeanFlow achieves comparable WER and speaker similarity results to the baseline, it experiences a drop in UTMOS scores. The speech generated by MeanFlow also exhibits noticeable background noise, likely due to the reduced batch size. This explains the drop in UTMOS despite the comparable WER and speaker similarity scores: the ASR and speaker verification models used are trained to be robust to noise, while UTMOS evaluates background noise as part of speech quality assessment. Generated speech samples can be found in the supplementary material.

In Table 3, Seed-TTS$_{SFT}$ refers to the Supervised Fine-Tuning task within the Seed-TTS framework, where the model is trained on labeled speech-text pairs to optimize text-to-speech performance metrics as described in Anastassiou et al. (2024). Compared to the 10-step Flow Matching baseline, our method reduces sampling steps by 5× and eliminates the need for Classifier-Free Guidance (CFG),

Table 3: Comparing SplitMeanFlow with Flow Matching and DMD on Seed-TTS$_{SFT}$ tasks.

| Method | NFE | CFG | WER (%, ↓) | SIM (↑) | CMOS (↑) |
|---|---|---|---|---|---|
| Flow Matching | 10 | Y | 5.51 | 0.79 | 0.00 |
| DMD | 2 | N | 5.61 | 0.79 | -0.04 |
| SplitMeanFlow | 2 | N | 5.61 | 0.79 | -0.01 |

significantly reducing computational overhead. Despite this acceleration, the quality degradation is minimal. The CMOS score of -0.01 indicates that human evaluators found the audio quality of our 2-step model almost identical to the 10-step baseline. We also compare our method to DMD Yin et al. (2024), another recent few-step generation method. SplitMeanFlow outperforms DMD in both speaker similarity and CMOS score, suggesting a perceptual preference for our model.

Table 4: Comparing SplitMeanFlow with Flow Matching on Seed-TTS$_{ICL}$ task.

| Model | NFE | CFG | WER (%, ↓) | SIM (↑) | CMOS (↑) |
|---|---|---|---|---|---|
| Flow Matching | 10 | Y | 2.86 | 0.69 | 0.00 |
| SplitMeanFlow | 2 | N | 2.97 | 0.68 | 0.00 |
| SplitMeanFlow | 1 | N | 2.86 | 0.69 | 0.00 |

We also evaluated SplitMeanFlow on Voice Cloning (Seed-TTS$_{ICL}$) tasks as detailed in Anastassiou et al. (2024). As shown in Table 4, the 2-step SplitMeanFlow model delivers strong performance, with a neutral CMOS score of 0, indicating that human evaluators found its quality equivalent to the 10-step baseline, despite minor fluctuations in objective metrics. Most notably, our 1-step Split-MeanFlow model achieves performance statistically comparable to the 10-step Flow Matching baseline across all metrics. The neutral CMOS score of 0 further confirms perceptual equivalence. This result demonstrates a 10-fold RTF acceleration and 20-fold reduction in computational cost without any noticeable quality loss, highlighting the effectiveness of learning the average velocity field via our algebraic consistency. This establishes SplitMeanFlow as a significant advancement toward truly one-step generative modeling.

Table 5: Comparing SplitMeanFlow with Flow Matching on Single Speaker SFT with DiTAR.

| Model | NFE | CFG | WER (%, ↓) | CMOS (↑) | RTF(↓) |
|---|---|---|---|---|---|
| Flow Matching | 10 | Y | 1.47 | 0.00 | 0.341 |
| SplitMeanFlow | 2 | N | 1.32 | -0.01 | 0.073 |
| SplitMeanFlow | 1 | N | 1.38 | -0.08 | 0.038 |

Single-speaker synthesis is a critical task in industrial TTS, where the goal is to achieve highly expressive and natural speech synthesis, possibly by sacrificing the voice cloning capacity. We evaluate SplitMeanFlow in this setup using the DiTAR framework. As shown in Table 5, even under the demanding conditions of single-speaker expressive speech generation, SplitMeanFlow exhibits only a minor CMOS degradation of 0.01. Remarkably, it achieves a 5-fold acceleration and a 10-fold reduction in computational cost, underscoring the practical applicability of SplitMeanFlow for large-scale, industrial-level TTS applications.

## 6 CONCLUSION

In this work, we introduced SplitMeanFlow, a novel framework for training few-step generative models. By returning to the first principles of average velocity and leveraging the additivity property of integrals, we derived the Interval Splitting Consistency objective that avoids the need for differential operators. We demonstrated that this algebraic formulation is a more general framework, with the differential identity in MeanFlow emerging as a limiting case. This results in significant practical advantages, including stable training, simpler implementation, and broader compatibility, as SplitMeanFlow eliminates the need for Jacobian-Vector Product (JVP) computations. We validate SplitMeanFlow on text-to-speech tasks with both AR and NAR setups, achieving 3 to 4-NFE inference on NAR and 1 to 2-NFE for AR models, without compromising speech generation quality. The algebraic perspective and practical effectiveness of SplitMeanFlow opens new possibilities for more efficient and powerful generative models.

## 7 REPRODUCIBILITY STATEMENT

The majority of our experiments are conducted using publicly available codebases (F5-TTS and CosyVoice2) and datasets (Emilia). We utilize publicly available checkpoints for model initialization and provide code snippets for implementing the initialization techniques in Appendix B, as well as for the boundary condition supervision in Appendix C. Readers should be able to easily reproduce our models based on these resources. Furthermore, we plan to release the code as pull requests to existing Flow Matching repositories to ensure compatibility with SplitMeanFlow, along with pre-trained models.

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

## A  PROOF: SPLITMEANFLOW AS A GENERALIZED CASE OF MEANFLOW

Our proof begins with the cornerstone of SplitMeanFlow, the Interval Splitting Consistency identity (Eq 10). To reveal its connection to MeanFlow, we first rearrange this equation algebraically:

$$\frac{(t-r)u(z_t, r, t) - (s-r)u(z_s, r, s)}{t-s} = u(z_t, s, t). \tag{15}$$

This form already resembles the definition of a derivative. We now investigate its behavior in the limit as the splitting point $s$ approaches the right endpoint $t$, i.e., $s \to t$.

1. **Analyzing the Right-Hand Side (RHS):** As $s \to t$, the length of the time interval $[s, t]$ for the average velocity $u(z_t, s, t)$ approaches zero. By its definition:

$$u(z_t, s, t) = \frac{1}{t-s} \int_s^t v(z_\tau, \tau) d\tau \tag{16}$$

which converges to the instantaneous velocity at that point. Thus:

$$\lim_{s \to t} u(z_t, s, t) = v(z_t, t). \tag{17}$$

2. **Analyzing the Left-Hand Side (LHS):** To clarify the structure of the LHS, we define an auxiliary function for the total displacement from $r$ to $t$, letting $g(t) = (t-r)u(z_t, r, t)$. The LHS of Eq. 15 can then be expressed as:

$$\frac{g(t) - g(s)}{t-s}. \tag{18}$$

This is precisely the definition of the derivative of the function $g$ at point $t$. Therefore:

$$\lim_{s \to t} \frac{g(t) - g(s)}{t-s} = g'(t). \tag{19}$$

3. **Connecting the Sides and Expanding:** By equating the limits of both sides, we arrive at the new identity $g'(t) = v(z_t, t)$. We now find the explicit form of $g'(t)$ by taking the total derivative of $g(t) = (t-r)u(z_t, r, t)$ with respect to $t$. Applying the product rule and the chain rule yields:

$$g'(t) = \frac{d}{dt}\left[(t-r)u(z_t, r, t)\right] = u(z_t, r, t) + (t-r)\frac{d}{dt}u(z_t, r, t). \tag{20}$$

Substituting this expansion back into $g'(t) = v(z_t, t)$, we get:

$$u(z_t, r, t) + (t-r)\frac{d}{dt}u(z_t, r, t) = v(z_t, t). \tag{21}$$

A simple rearrangement recovers the core differential identity of MeanFlow:

$$u(z_t, r, t) = v(z_t, t) - (t-r)\frac{d}{dt}u(z_t, r, t). \tag{22}$$

## B  INITIALIZATION STRATEGY

SplitMeanFlow models the average velocity between timesteps $t$ and $r$. Both timesteps are passed through the same embedding network, concatenated, and then projected back into the feature space of $t$ via a linear mapping $\mathbf{W}$.

Let the embeddings for $t$ and $r$ be denoted as $\mathbf{e}_t = \mathcal{E}(t)$ and $\mathbf{e}_r = \mathcal{E}(r)$, respectively. The concatenated embeddings $\mathbf{e}_{t,r}$ are projected to $\mathbf{e}'_{t,r}$ as follows:

$$\mathbf{e}_{t,r} = [\mathbf{e}_t, \mathbf{e}_r]; \quad \mathbf{e}'_{t,r} = \mathbf{W}\mathbf{e}_{t,r} \tag{23}$$

To ensure that SplitMeanFlow retains the behavior of the original flow-matching model, the linear mapping $\mathbf{W}$ is initialized as:

$$\mathbf{W} = [D_{diag} \quad 0] \tag{24}$$

where $D_{diag}$ represents a diagonal matrix. This initialization guarantees that the model behaves identical to the original model at the start of training. The code snippet is shown as following:

```
self.time_proj = nn.Linear(dim * 2, dim)
with torch.no_grad():
    self.time_proj.weight.zero_()
    self.time_proj.bias.zero_()
    self.time_proj.weight[:, :dim] = torch.eye(dim)
```

## C  TEACHER'S VELOCITY AS THE TRAINING ANCHOR

### C.1  METHODOLOGY

In a distillation setup, we observed that using the instantaneous velocity of the teacher model as an anchor to prevent training collapse can lead to a performance drop in both objective and subjective metrics. To address this, we propose using the approximated average velocity of the teacher over the interval $[t, r]$ as the anchor. This is achieved by performing iterative sampling of the teacher model during distillation. The interval $[t, r]$ is discretized into $n$ subintervals, with time steps $t_0 = t, t_1, \ldots, t_n = r$. At each step, the teacher model evolves the state based on the discrete approximation of the ODE:

$$z_{t_{k+1}} = z_{t_k} + (t_{k+1} - t_k) \cdot v(z_{t_k}, t_k; \theta) \tag{25}$$

where $t_0 = t$, $t_n = r$, and $t_1, t_2, \ldots, t_{n-1}$ are intermediate time steps. The total displacement over the interval $[t, r]$ is computed as:

$$\Delta z^{\text{teacher}} = \sum_{k=0}^{n-1} \left(z_{t_{k+1}} - z_{t_k}\right) = \sum_{k=0}^{n-1}(t_{k+1} - t_k) \cdot v(z_{t_k}, t_k; \theta) \tag{26}$$

This discrete displacement approximates the integral of the instantaneous velocity $v(z_t, t; \theta)$ over $[t, r]$, which is the continuous process the student model aims to replicate. To approximate the average velocity, we normalize the displacement by the length of the interval:

$$\bar{v}_{\text{teacher}}(z_t, t, r) = \frac{\Delta z^{\text{teacher}}}{r - t} \tag{27}$$

Thus, the teacher's discrete displacement provides a numerical approximation of this integral. The anchor loss is then defined as:

$$L_{\text{anchor}} = \mathbb{E}_{t,r} \left[\left\|u_{\text{student}}(z_t, t, r) - \bar{v}_{\text{teacher}}(z_t, t, r)\right\|^2\right] \tag{28}$$

It is important to note that this strategy may impact training speed due to the multiple NFEs required by the teacher model during distillation. In our experiments, we set the teacher NFE to 2 during distillation.

### C.2  IMPLEMENTATION

```
@torch.no_grad()
def teacher_average_velocity(z, cond, text, mask, t, r, teacher,
    teacher_nfe, teacher_cfg):
    z0 = z
    t_vals = get_epss_timesteps(teacher_nfe)
    t_vals = t_vals - (torch.cos(torch.pi / 2 * t_vals) - 1 + t_vals)
    t_vals = t_vals[None, :] * (r - t)[:, None] + t[:, None]
    for i in range(0, teacher_nfe):
        t_, r_ = t_vals[:, i], t_vals[:, i + 1]
        v = teacher(x=z, cond=cond, text=text, time=t_, mask=mask)
        pred, null_pred = torch.chunk(v, 2, dim=0)
        v = teacher_cfg * pred + (1 - teacher_cfg) * null_pred
```

```
        z = z + (r_ - t_)[:, None, None] * v
    v_target = (z - z0) / (r - t)[:, None, None]
    return v_target

def get_epss_timesteps(n):
    dt = 1 / 32
    predefined_timesteps = {
        2: [0, 10, 32],
        3: [0, 4, 12, 32],
        4: [0, 2, 6, 12, 32],
        5: [0, 2, 4, 8, 16, 32],
        6: [0, 2, 4, 6, 8, 16, 32],
        7: [0, 2, 4, 6, 8, 16, 24, 32],
        8: [0, 2, 4, 6, 8, 16, 24, 28, 32],
        10: [0, 2, 4, 6, 8, 12, 16, 20, 24, 28, 32],
        12: [0, 2, 4, 6, 8, 10, 12, 14, 16, 20, 24, 28, 32],
        16: [0, 1, 2, 3, 4, 5, 6, 7, 8, 10, 12, 14, 16, 20, 24, 28, 32],
    }
    if n not in predefined_timesteps:
        return torch.linspace(0, 1, n + 1)
    return dt * torch.tensor(predefined_timesteps[n])
```

## C.3 ABLATION STUDY

Table 6 compares the performance of using the teacher's instantaneous velocity versus the approximated average velocity as the training anchor for SplitMeanFlow on F5-TTS. The results show a clear degradation in all metrics. This suggests that the instantaneous velocity may not provide an optimal anchor for training.

Table 6: Comparing anchor with teacher's instantaneous velocity and average velocity.

| Training Anchor | WER ($\%, \downarrow$) | SIM ($\uparrow$) | UTMOS($\uparrow$) | CMOS($\uparrow$) |
|---|---|---|---|---|
| **Seed-TTS *test-en*** | | | | |
| teacher approximated average velocity | 1.67 | 0.65 | 3.72 | -0.03 |
| teacher instantaneous velocity | 2.07 | 0.59 | 3.61 | -0.08 |
| **Seed-TTS *test-zh*** | | | | |
| teacher approximated average velocity | 1.66 | 0.74 | 3.01 | -0.01 |
| teacher instantaneous velocity | 2.04 | 0.65 | 2.92 | -0.11 |

## D SPLITMEANFLOW IN NAR TTS: DETAILED RESULTS

### D.1 SPEECH GENERATION QUALITY VS. FLOW RATIO

The Flow Ratio reference to the $p$ in section 5.1.

Table 7 presents an ablation study on the flow ratio $p$ for SplitMeanFlow on F5-TTS. When $p \geq 0.5$, the results show comparable performance across the objective metrics, indicating that the model is correctly guided by the supervision of boundary conditions. However, when $p = 0.3$, both speaker similarity and UTMOS experience noticeable degradation. This suggests that the boundary condition anchor becomes less effective at this ratio, leading to a deterioration in the quality of the generated speech. At $p = 0.1$, the model fails to converge, with training collapsing entirely. This indicates that the anchor is no longer providing meaningful guidance, and the training process is unable to proceed in a reasonable direction.

Table 7: Ablation on flow ratio for SplitMeanFlow on F5-TTS.

| Flow Ratio | WER (%, ↓) | SIM (↑) | UTMOS(↑) |
|---|---|---|---|
| **Seed-TTS *test-en*** | | | |
| 0.1 | 103.75 | 0.05 | 1.34 |
| 0.3 | 1.58 | 0.59 | 3.60 |
| 0.5 | 1.67 | 0.65 | 3.72 |
| 0.7 | 1.71 | 0.64 | 3.75 |
| 0.9 | 1.65 | 0.64 | 3.65 |
| **Seed-TTS *test-zh*** | | | |
| 0.1 | 122.41 | 0.06 | 1.23 |
| 0.3 | 1.73 | 0.68 | 2.87 |
| 0.5 | 1.66 | 0.74 | 3.01 |
| 0.7 | 1.68 | 0.74 | 2.98 |
| 0.9 | 1.62 | 0.73 | 3.01 |

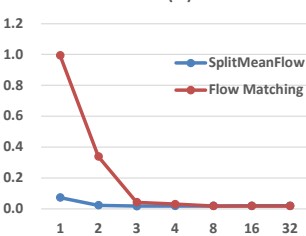 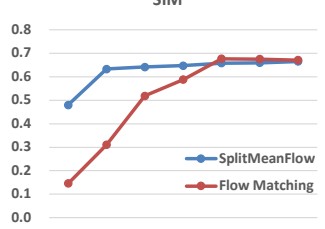 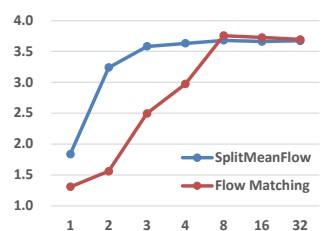

Figure 2: Comparison of speech quality metrics between Flow Matching and SplitMeanFlow as a function of inference NFE.

## D.2 SPEECH GENERATION QUALITY VS INFERENCE NFE

Figure 2 illustrates the speech generation quality metrics as a function of inference NFE for SplitMeanFlow and Flow Matching on F5-TTS. The results clearly show that SplitMeanFlow outperforms the Flow Matching baseline at lower NFEs. As the NFE increases, the performance gap between the two approaches narrows. When the NFE approaches 32, both methods exhibit degradation in objective metrics. This behavior reflects a typical challenge in TTS evaluation, where objective metrics no longer effectively capture improvements in generation quality, making CMOS evaluation necessary at this point.

## E TRAINING DETAILS

For F5-TTS, we use the official Base model with 336M parameters as the Flow Matching teacher. During distillation, we employ the Adam Kingma & Ba (2015) optimizer with a learning rate of $3 \times 10^{-5}$. We initialize with the official F5-TTS Base checkpoint and continue training using the distillation strategy for 300K steps.

For CosyVoice2, we use the official flow checkpoint with 107M parameters. The distillation process also uses the Adam optimizer with a learning rate of $3 \times 10^{-5}$. We initialize with the official CosyVoice2-0.5B flow module checkpoint and continue training with the distillation strategy for 200K steps.

For the single-speaker configuration of DiTAR, we first pretrain the model on a large-scale internal multi-speaker corpus and then fine-tune it on a small single-speaker dataset. During SplitMeanFlow distillation, we update only the LocDiT component of DiTAR Jia et al. (2025), using the single-speaker data. We distill LocDiT for 100K steps with a learning rate of $1 \times 10^{-5}$.

## F   THE USE OF LARGE LANGUAGE MODELS

We acknowledge the use of LLM for assisting with tasks such as grammar correction, enhancing expression diversity, formatting tables, and debugging code. However, we emphasize that all core ideas, experiments, and analyses are original contributions.

