# OpenReview forum: "SplitMeanFlow: Interval Splitting Consistency in Few-Step Generative Modeling"
_ICLR.cc/2026/Conference — Submitted to ICLR 2026_

### Official Review · Reviewer_3Vzm · 2025-10-25

**Soundness:** 3
**Presentation:** 2
**Contribution:** 2
**Rating:** 2
**Confidence:** 4

**Summary:**

This paper proposes SplitMeanFlow, a variant of MeanFlow built on interval-splitting consistency. Unlike MeanFlow, which is derived from a differential identity, SplitMeanFlow is formulated from a simple algebraic identity underlying flow-matching trajectories. This formulation eliminates the need for JVP operations during training. Furthermore, SplitMeanFlow employs the original flow-matching objective as a boundary condition to prevent performance degradation. In practice, SplitMeanFlow adopts a two-stage training procedure: (1) the backbone model is first trained via flow matching to serve both as a ground-truth target for the boundary condition and as initialization for SplitMeanFlow training; (2) SplitMeanFlow is then trained to refine the model. The authors apply the proposed method to TTS tasks, and show that it can generate high-quality speech with fewer generation steps.

**Strengths:**

- The proposed method is intuitive and easy to implement.
- It further reduces generation steps compared to pretrained flow-matching models, demonstrating improved efficiency.

**Weaknesses:**

1. The conceptual advancement over existing methods seems limited. For example, 'One-Step Diffusion via Shortcut Models'[1] also leverages self-consistency within flow-matching trajectories. Although the authors note that shortcut models rely on discrete time intervals with a restricted formulation s=(r+t)/2, the paper does not clearly demonstrate the advantages of the proposed continuous formulation. Further theoretical justification or empirical comparison would be valuable.
2. SplitMeanFlow appears to be a general approach applicable to various synthesis tasks. However, experiments are limited to TTS. Demonstrating its effectiveness on other modalities (e.g., image or video generation) would strengthen the validity of the proposed method.
3. While JVPs are fundamental operations well supported by modern deep learning frameworks, the paper does not clearly articulate when or why avoiding them is beneficial. The authors should provide concrete evidence or benchmarks showing the computational or practical advantages (e.g., reduced overhead, improved efficiency, or unsupported hardware scenarios).
4. Section 4.3 presents the mathematical relationship between MeanFlow and SplitMeanFlow. For me, however, it’s rather straightforward and unnecessary to be included in the main text. Instead, a deeper analysis of the relationship between SplitMeanFlow and consistency models would enrich the discussion, as both share the common goal of reducing inference steps through consistency.
5. Metrics such as UTMOS and CMOS should be reported with confidence intervals to support claims of significance.
6. Details about the human evaluation setup (e.g., number of participants, number of evaluated samples, and protocol) are insufficient and should be clarified.
7. In the experiments, CFG is not applied to SplitMeanFlow, which appears to improve the RTF and reduce computational cost. However, this improvement is achievable when the target is generated using CFG from the teacher model during the second stage. As this setup is not an inherent feature of SplitMeanFlow, the reported gains in RTF and computational cost are somewhat overstated and may not reflect the true efficiency of the method. A similar approach is already proposed in 'Towards Flow-Matching-based TTS
without Classifier-Free Guidance'[2].

[1] Frans, K., Hafner, D., Levine, S. and Abbeel, P., One Step Diffusion via Shortcut Models. In The Thirteenth International Conference on Learning Representations.

[2] Liang, Y., Liu, W., Qiang, C., Niu, Z., Chen, Y., Ma, Z., Chen, W., Li, N., Zhang, C. and Chen, X., 2025. Towards Flow-Matching-based TTS without Classifier-Free Guidance. arXiv preprint arXiv:2504.20334.

**Questions:**

- Why are the MeanFlow results for NFEs of 3 and 4 not presented in Table 1?
- Why are the MeanFlow results for NFEs of 2 not presented in Table 2?
- Why was TTS chosen as the main experimental task? If SplitMeanFlow is not specifically tailored to TTS, evaluating it on a broader range of synthesis tasks would strengthen its effectiveness.

**Details Of Ethics Concerns:**

The description of the human evaluation procedure is not clear. More detailed explanation for the subjective evaluation procedures can be added.

---

### Official Review · Reviewer_hvLe · 2025-10-26

**Soundness:** 3
**Presentation:** 3
**Contribution:** 2
**Rating:** 4
**Confidence:** 3

**Summary:**

The paper proposes SplitMeanFlow, a few-step generative modeling framework that learns an average-velocity field by enforcing Interval Splitting Consistency identity over time intervals. Concretely, the key constraint is $(t-r)u(z_t,r,t)=(s-r)u(z_s,r,s)+(t-s)u(z_t,s,t)$, which yields a self-consistency objective and avoids Jacobian–vector products. The authors show that MeanFlow’s differential identity is recovered in the $s\to t$ limit. They apply the method on text-to-speech tasks, reporting 1–4 NFEs at inference with comparable WER/SIM/CMOS to multi-step Flow Matching baselines and large RTF gains with ~20× compute reduction in some settings.

**Strengths:**

- The core identity comes directly from integral additivity. It's conceptually clear. The derivation that MeanFlow is the limiting special case gives the method theoretical connection to prior works.
- Comparing to MeanFLow, SplitMeanFlow avoids JVP calculation, lowering memory consumption.
- Relatively strong empirical results in TTS. Achieving 1-4 NFEs with quality at parity with 32-step Flow Matching.
- Has a clear reproducibility statement. Experiments leverage public codebases (F5-TTS, CosyVoice2) and datasets (Emilia).

**Weaknesses:**

- All main experiments are in TTS. While compelling, it is unclear how well the identity and training dynamics transfer to image generation where evaluation and inductive biases differ.
- Reliance on distillation teacher supervision. The strongest results come from a Flow-Matching teacher → SplitMeanFlow student procedure. Also, the current training need to use the teacher’s velocity $v(z_t, t)$ to ensure boundary conditions, avoiding degenerate solutions. Combined with comparable performance to other flow-matching methods in the main experiment, this raises the question of whether this method and its future extensions has the potential to yield performance much beyond the SOTA performance.
- The baselines include Flow Matching, MeanFlow, and DMD. Broader comparisons to SOTA TTS methods with the same setups would strengthen claims about empirical advantages.

**Questions:**

- Most flow-matching works demonstrate their empirical performance on image generation tasks. What's the intuition behind evaluating on TTS tasks in this paper? Have you tried standard image benchmarks to demonstrate generality? What adaptations, if any, were missing to make it work?
- The appendix shows collapse at low flow-ratio. Do you have potential solutions to fix this or to get rid of the teacher velocity mix-in during training?

---

### Official Review · Reviewer_mxr3 · 2025-10-27

**Soundness:** 2
**Presentation:** 2
**Contribution:** 2
**Rating:** 4
**Confidence:** 4

**Summary:**

The paper proposes SplitMeanFlow, an algebraic framework for few‑step/one‑step generative modeling that learns the average velocity field by enforcing a new identity called Interval Splitting Consistency, which is a more general framework of MeanFlow.
Using SplitMeanFlow objective eliminates Jacobian-vector products (JVP) operation during training.
Text-to-speech (TTS) experiments demonstrate that SplitMeanFlow can one-step/few-step speech synthesis.

**Strengths:**

- SplitMeanFlow replaces the differential MeanFlow identity with a purely algebraic self-consistency derived from additivity of definite integrals: $(t-r)u(z_t,r,t) =(s-r)u(z_s,r,s)+(t-s)u(z_t,s,t)$. This algebraic constraint removes JVP operation and gives a simple, theoretically grounded training objective for learning the average-velocity field.

- Another strength is that TTS experiments show that SplitMeanFlow can one-step/few-step generation on top of the multiple different architectures/models.

**Weaknesses:**

The major limitation of this work as a scientific paper is its experimental evaluation.

Specifically, the experimantal validation is limited only to TTS even though, excluding the abstract, this work rarely mention about speech from the introduction until the experiments through the related work. The sudden focus on TTS-only evaluation in the experiments feels abrupt, and even within this evaluation, the comparison against existing diffusion-based, flow-matching-based, and their accelerated TTS methods is insufficient [1-5].

- If this work is positioned as a new acceleration methods for flow-matching-based TTS, the introduction and related work should thoroughly discuss existing TTS methods, clearly motivate the paper's contribution in that context, and select appropriate baseline models for comparison in the experiments.

- If this work is positioned as a general extension to generative models, the current TTS experiments would be suitable for an appendix. However, the main paper should include experiments on standard benchmarks (e.g., CIFAR-10, ImageNet) and compare against relevant prior work, e.g., [6-8]. In this case, it would also be desirable to thoroughly discuss closely related works, such as Consistency Trajectory Models (CTM) [9], Align Your Flow [10], and Flow Map Matching [11], in the main body. (Furthermore, this would imply that the primary area for submission should have been 'generative modeling'.)


Additionally, the claims regarding computational efficiency and stable training from eliminating JVP, stated as a main contribution in the introduction, require a more detailed evaluation.
- Regarding computational efficiency, rather than just stating it in a table caption, the paper should specify, for example, the GPU and more detailed experimental setup and provide concrete metrics (e.g., GPU memory usages, training speed in GPU hours or trainig iterations) in comparison to the MeanFlow.
- Similarly, for training stability, a discussion using training/validation curves under more detailed experimental conditions would be more convincing. The reviewer is particularly concerned that the training stability may not stem from the SplitMeanFlow formulation itself, given that Appendix C.1 appears to perform a distillation process similar to CTM/SoundCTM[12].

[1] Ye, Zhen, et al. "Comospeech: One-step speech and singing voice synthesis via consistency model." ACMMM 2023.

[2] Ye, Zhen, et al. "Flashspeech: Efficient zero-shot speech synthesis." ACMMM 2024.

[3] Guan, Wenhao, et al. "Reflow-tts: A rectified flow model for high-fidelity text-to-speech." ICASSP 2024.

[4] Wang, Kaidi, et al. "Slimspeech: Lightweight and efficient text-to-speech with slim rectified flow." ICASSP 2025.

[5] Park, Hyun Joon, et al. "RapFlow-TTS: Rapid and High-Fidelity Text-to-Speech with Improved Consistency Flow Matching." Interspeech 2025.

[6] Lu, Cheng, et al. "Simplifying, Stabilizing and Scaling Continuous-Time Consistency Models." ICLR 2025.

[7] Zhou, Linqi, et al. "Inductive moment matching." ICML 2025.

[8] Frans, Kevin, et al. "One step diffusion via shortcut models." ICLR 2025.

[9] Kim, Dongjun, et al. "Consistency trajectory models: Learning probability flow ode trajectory of diffusion." ICLR 2024.

[10] Sabour, Amirmojtaba, et al. "Align Your Flow: Scaling Continuous-Time Flow Map Distillation." NeurIPS 2025.

[11] Boffi, Nicholas Matthew, et al. "How to build a consistency model: Learning flow maps via self-distillation." NeurIPS 2025.

[12] Saito, Koichi, et al. "SoundCTM: Unifying Score-based and Consistency Models for Full-band Text-to-Sound Generation." ICLR 2025.

**Questions:**

- Is there a specific reason for limiting the experiments to only TTS?
- Given the tight connection between MeanFlow/SplitMeanFlow and CTM, and considering that Appendix C.1 seems to practically distill a teacher's ODE trajectory via time discretization even within the SplitMeanFlow formulation, could you explain the practical differences between the SplitMeanFlow and CTM/SoundCTM? (The authors could check the theoretical connection in Appendix D&E in [10] and Appendix A in [13], for example)

[13] Hu, Zheyuan, et al. "CMT: Mid-Training for Efficient Learning of Consistency, Mean Flow, and Flow Map Models." arXiv preprint arXiv:2509.24526 (2025).

---

### Official Review · Reviewer_rRik · 2025-10-31

**Soundness:** 3
**Presentation:** 3
**Contribution:** 2
**Rating:** 6
**Confidence:** 3

**Summary:**

This paper introduces SplitMeanFlow, a framework for few-step generative modeling that reformulates average velocity field learning through an algebraic identity termed "Interval Splitting Consistency," derived from the additivity property of definite integrals. Unlike MeanFlow's differential formulation that requires computationally expensive Jacobian-vector products (JVP), SplitMeanFlow enforces consistency across interval decompositions algebraically, with the authors proving that MeanFlow's differential identity emerges as a limiting case when the splitting point approaches the interval endpoint. The method is validated on large-scale text-to-speech synthesis across multiple architectures (F5-TTS, CosyVoice2, DiTAR, Seed-TTS), demonstrating 3-4 step inference for non-autoregressive models and 1-2 steps for autoregressive models, achieving 10× speedup and 20× computational reduction while maintaining speech quality comparable to 32-step flow matching baselines.

**Strengths:**

1. Strong theoretical foundation with elegant algebraic formulation. The paper provides a rigorous mathematical derivation of the Interval Splitting Consistency identity from first principles (Equations 8-11), showing how the additivity property of integrals naturally leads to an algebraic constraint on average velocity fields. The theoretical analysis in Appendix A convincingly demonstrates that MeanFlow's differential identity (Equation 7) emerges as a special limiting case, establishing SplitMeanFlow as a more general framework that avoids infinitesimal approximations.
2. Significant practical advantages with comprehensive validation. The elimination of JVP computations addresses a major practical limitation of MeanFlow, as evidenced by the authors' observation that "training F5-TTS with MeanFlow is not feasible due to excessive memory consumption from JVP calculations, which forces the batch size to 1" (page 8, lines 394-395). The experimental validation spans multiple architectures (F5-TTS, CosyVoice2, DiTAR, Seed-TTS) across both autoregressive and non-autoregressive settings, demonstrating robust performance improvements with minimal quality degradation. The detailed ablation studies on flow ratio (Table 7) and inference NFE (Figure 2) provide valuable insights into the method's behavior.

**Weaknesses:**

1. Missing key comparisons weaken empirical contributions. First, the paper lacks direct comparison with the shortcut model (Frans et al., 2025) despite acknowledging its partial equivalence to SplitMeanFlow for the special case s=(r+t)/2 (lines 307-311). The dismissal that "design philosophies differ significantly" is inadequate without empirical comparison showing performance differences. Second, Table 2 (page 8) reports MeanFlow results with reduced batch size (0.1× original) to "accommodate memory consumption" (lines 421-422), but the comparison is unfair as batch size significantly affects training dynamics and final performance.

2. Theoretical analysis lacks depth in several important aspects. While the algebraic derivation is sound, several theoretical questions remain unaddressed. First, the paper does not provide convergence guarantees or sample complexity analysis for the SplitMeanFlow training objective. The statement that "boundary conditions are required to avoid collapsing to a trivial solution" (lines 365-366) is mentioned but not formally analyzed—what are the specific degeneracies that can occur, and under what conditions does the method provably converge to the correct average velocity field?

3. Limited evaluation to a single domain restricts generalizability assessment. All experiments are conducted exclusively on text-to-speech synthesis tasks, leaving the method's effectiveness on other modalities unexplored. The paper claims in the abstract that Flow Matching has achieved "prominent performance in generative modeling" across "images, video, and audio" (lines 12-14), yet only validates SplitMeanFlow on speech. Given that different modalities have distinct characteristics (e.g., images have spatial structure, video has temporal dynamics), demonstrating effectiveness on at least image generation would significantly strengthen the claims.

4. Minor Issues. The paper focuses on average velocity fields but does not discuss recent theoretical advances in higher-order flow matching that directly relate to your framework. [1] develops a unified framework for high-order flow matching with sharp statistical convergence rates, which could contextualize the theoretical properties of learning average velocities over intervals. Similarly, [4] provides theoretical analysis of high-order mean flow models including expressivity guarantees and provably efficient criteria, which seems highly relevant given that SplitMeanFlow learns mean velocities over arbitrary intervals [r,t]. [2] explores noise-robust generative modeling through high-order mechanisms, and [3] examines physics-inspired constraints for stable flow matching, both of which relate to training stability concerns you address through boundary conditions in Section 5.1. Additionally, [5] provides theoretical analysis of discrete flow matching, which could offer insights into the discretization of your continuous interval splitting formulation used during training.

# Reference

[1] Maojiang Su, Jerry Yao-Chieh Hu, Yi-Chen Lee, Ning Zhu, Jui-Hui Chung, Shang Wu, Zhao Song, Minshuo Chen, and Han Liu: High-order flow matching: Unified framework and sharp statistical rates. NeurIPS 2025.

[2] Bo Chen, Chengyue Gong, Xiaoyu Li, Yingyu Liang, Zhizhou Sha, Zhenmei Shi, Zhao Song, Mingda Wan, Xugang Ye:NRFlow: Towards Noise-Robust Generative Modeling via High-Order Mechanism. UAI 2025.

[3] Yang Cao, Bo Chen, Xiaoyu Li, Yingyu Liang, Zhizhou Sha, Zhenmei Shi, Zhao Song, Mingda Wan: Force Matching with Relativistic Constraints: A Physics-Inspired Approach to Stable and Efficient Generative Modeling. CIKM 2025.

[4] Yang Cao, Yubin Chen, Zhao Song, Jiahao Zhang: Towards High-Order Mean Flow Generative Models: Feasibility, Expressivity, and Provably Efficient Criteria. arXiv:2508.07102.

[5] Maojiang Su, Mingcheng Lu, Jerry Yao-Chieh Hu, Shang Wu, Zhao Song, Alex Reneau, Han Liu: A Theoretical Analysis of Discrete Flow Matching Generative Models. arXiv:2509.22623.

**Questions:**

1. Could you conduct experiments comparing SplitMeanFlow against the shortcut model on at least one benchmark (e.g., F5-TTS or CosyVoice2) under identical training conditions, reporting WER, speaker similarity, UTMOS, and computational metrics?

2. Is it possible to train MeanFlow with gradient accumulation to match SplitMeanFlow's effective batch size, or report results for both methods across multiple batch sizes to isolate the contribution of your algebraic consistency objective?

3. Is it possible to demonstrate SplitMeanFlow's effectiveness on at least one additional modality, such as image generation on ImageNet or CIFAR-10?

---

### Meta-Review · Area_Chair_5eM5 · 2026-01-03

**Summary:**

Primary concern of reviewers is that this work is posed as a general improvement to flow matching, but provides results only on text-to-speech tasks, which is not the convention for this type of work. As a result, this work does not include sufficient comparisons to closely related mean-flow improvement works.

**Reviewer Concerns:**

Authors did not provide a rebuttal

**Reviewer Scores:**

N/A as authors did not provide a rebuttal

---

### Decision · Program_Chairs · 2026-01-26

Reject